# Higher Education Institution beyond the COVID-19 Pandemic—Evidence from Romania

**Sofia David [1], Ludmila Daniela Manea [2], Florina Oana Virlanuta [2,\*], Nicoleta Bărbuță-Mișu [1] and Iulian Adrian Șorcaru [2]**

1   Department of Business Administration, "Dunarea de Jos" University of Galati, 800008 Galati, Romania
2   Department of Economics, "Dunarea de Jos" University of Galati, 800008 Galati, Romania
\*   Correspondence: florina.virlanuta@ugal.ro

**Abstract:** The COVID-19 pandemic has profoundly affected the education system in Romania. The major objective of this study was to identify the challenges in higher education institutions beyond the COVID-19 pandemic. Our study is based on a questionnaire-type analysis with 388 respondents (students from different universities). Using the SEM-PLS method, we designed a conceptual model, which is based on seven latent variables: a resilient education system in the context of the COVID-19 pandemic (SER); teacher–student, student–teacher, and student–institution communication (PS); logistical assistance from the educational institution (AL); adaptation according to knowledge-economy requirements (EC); online teaching–learning in higher education (API); a hybrid education model (EH); and digital skills and the integration of digital technology in institutions (ITE). We formulated seven hypotheses in order to test the strength of the correlation between the latent variables. Our research highlights a significant correlation between logistical assistance from the educational institution (LA) and teacher–student, student–teacher, and student–institution (PS) communication. Moreover, logistical assistance from the educational institution (LA) has a significant effect on the teaching–learning activity.

**Keywords:** COVID-19 pandemic; resilient education; hybrid education model; knowledge economy; Romania

## 1. Introduction

The COVID-19 pandemic has profoundly affected education and exacerbated existing social inequities by closing schools in Europe and Central Asia. Ensuring the continuity of education during the pandemic has proved to be a difficult task worldwide.

Romania is among the countries where pre-university education has been most affected by the pandemic, the effects being much more visible in comparison to university education. According to the official statistics provided by the Ministry of Education, in Romania, 65,000 children were left out of school hours because they did not have the necessary resources to access online education. However, in Romanian universities, the student' dropout rate decreased during 2020–2021.

During the COVID-19 pandemic, all of society faced unexpected difficulties. In previous studies, these difficulties were indicated as "grand challenges", characterized by their complexity [1,2]. Thus, for educational institutions, a solution to the difficulties that were specific to traditional learning was electronic learning.

Globally, the COVID-19 pandemic has had a particularly negative impact on the mobility of international students. Therefore, the chance for them to attend full-time courses abroad or experience exchange opportunities was no longer possible [3]. However, the opportunity to participate in virtual mobility emerged, which in turn offered the opportunity to attend courses delivered by different institutions around the world, being able to choose the most successful ones for each discipline. Moreover, the mobility of

teachers decreased considerably during this period. In order to reduce the effects of this situation, many webinars were organized, allowing students to participate in online education offered by professors from abroad.

University studies are an important investment of time and resources for students, their families, and society. Therefore, ensuring a high degree of graduates is a fundamental responsibility of all universities [4]. However, the rapid development of all economic sectors makes it very difficult to predict what types of jobs will be available in the future [4]. There are many studies that address the effects of the pandemic on education [5,6], with some authors pointing out that social distancing is an inhibitory factor in the cognitive development of pupils and students [7,8]. Others highlight changes in motivation and student satisfaction in the e-learning system [9]. Although there has been a difficult period of adaptation to online education for both teachers and students, the latter prefer to continue their online learning activities. Some researchers have focused on the effects of the pandemic context on students' mental health. Thus, as a result of the studies based on the questionnaire, difficulty concentrating and stress caused by concerns for health and family were identified as effects of the COVID-19 pandemic [10,11].

In this regard, the research questions in our study are the following:

**RQ1:** What are the challenges in higher education institutions beyond the COVID-19 pandemic, according to students' perceptions?

**RQ2:** What are the students' perspectives on the hybrid education model?

**RQ3:** What is the relationship between the latent variables involved in the research?

## 2. Literature Review

The transition from online to onsite education was very slow in the higher education system. Beginning in November 2021, one by one, the Romanian universities started to recall their students. Each institution decided, based on its decision-making autonomy, when and how the onsite lectures would take place. Moreover, the faculties within the Romanian universities, depending on the number of students in each class, were given the opportunity to decide how lectures should take place. The field of study was also a very important determinant for such a decision. For example, in some fields of study, e.g., economics, law, etc., where laboratories/practical lessons are less utilized, the online and hybrid education systems were used for a larger period. Therefore, for some faculties, the transition from online to onsite education was very slow, especially for private universities compared to state universities.

During the COVID-19 pandemic, we saw a growing research interest in the instructional strategies of online teaching [12], teaching platforms [13], educational resources, practices, and strategies [14,15], and the COVID-19 lockdown's impact on student learning [16]. In response to the coronavirus lockdown, remote learning was the only solution for the education sector in order to minimize the impact on the academic progression of students [17,18].

At the same time, the number of studies approaching the transition from onsite to online education during the COVID-19 lockdown expanded significantly. The first studies highlighted the negative impacts of COVID-19 on education, while the most recent studies highlighted its positive impact or advantages of it. This led to the idea that online education might be a part of future education [19,20].

A large-scale study was conducted by Aristovnik et al. [21] on a sample of 30,383 students from 62 countries (including Romania) and six continents in order to determine the impacts of the COVID-19 pandemic on the life of higher education students during the first wave of the pandemic, such as academic work and life, social life, change in habits, emotional life, personal circumstances, the role of institutions, measures taken by institutions, and personal reflections. The empirical results of this research revealed that females, full-time students, postgraduate students, and social science students were mainly less affected by the COVID-19 pandemic. Furthermore, students with a better standard of living and students from Oceania and Europe showed a more positive attitude to most

aspects of student life. The study also revealed that teaching and support staff played an important role in maintaining students' satisfaction with the university.

Another study on 307 Agricultural students from India, which analyzed the students' preferences and perceptions regarding online classes, found a positive attitude of the students towards online classes and revealed some benefits of online learning, such as flexibility and convenience for the learners. The results of the study also highlighted the factors that could lead to the failure of online classes, such as technological constraints, distractions, instructors' incompetency, learners' inefficacy, and health issues. The respondents preferred well-structured content with recorded videos and interactive sessions with quizzes and assignments [22].

A survey based on 216 tourism and hospitality students in Macau, conducted by Agyeiwaah et al. [23], outlined that three online learning attributes, as perspicuity and dependability, stimulation and attractiveness, and usability and innovation, significantly impacted students 'satisfaction with online learning.

The findings of a study conducted by Ismaili in May 2020 on 108 students from Eotvos Lorand University (ELTE) in Budapest, Hungary, showed positive attitudes and willingness of the students to engage in distance learning in the post-COVID19 pandemic, which means there is an immense future potential for e-learning platforms in higher education institutions [24].

A study conducted by Almossa between March and May 2020 in Saudi Arabia explored the students' perspectives towards learning and assessment using quantitative and qualitative tools for analyzing Twitter data. The findings indicated that student engagement was affected due to the challenges of online learning and assessment. The study also identified some factors that influenced students' engagement with learning and assessment, such as communication, fairness, and technical and assessment issues [25].

In a study of 179 students from Saudi Arabia, Abumalloh et al. examined the expected benefits of e-learning during the COVID-19 pandemic. Based on the Push–Pull–Mooring theory, the findings revealed that the push factor (environmental threat), the pull factors (e-learning motivation, perceived information sharing, and social distancing), and one mooring factor (perceived security) significantly impact learners' benefits [26].

Tang made a comparative analysis of students' live online learning readiness during the coronavirus pandemic in higher education. The study revealed no significant differences between male and female students, but there were significant differences between students from different levels of studies regarding motivation for learning, learner control, and self-directing learning ability [27].

Based on a qualitative methodology, a study on pharmacy students from the Kingdom of Saudi Arabia conducted by Ali [28] highlighted some facilitators (easier and more frequent communication with the academic staff and communication between the learners) and barriers (technology problems, inappropriate teaching, assessment methods, and limitation of the technology) which affected students' education during the lockdown. The respondents made suggestions for improving online education, such as the provision of recorded lectures and the need for academic staff to modify their teaching methods. Regarding the long-term impact of online education during the lockdown, the respondents highlighted improved grade point average (GPA), the skill that they had learned (e.g., multitasking in a short time, academic writing, time management, working under pressure), and the skills students could have learned better in onsite education (e.g., practical skills, oral communication, team working, presentation skills, group communication). Moreover, the respondents viewed the future of pharmacy education as a hybrid of online and onsite learning.

Butnaru et al. [29] conducted a study on 665 bachelor's degree and master's degree students from Romanian universities that analyzed the perceptions of Romanian students regarding the effects of online education during the COVID-19 pandemic from the perspective of their wellbeing. The results of the study show a negative relationship between students' desire to study onsite and their wellbeing; confirmed a statistically significant

effect of the negative perception of personal development on students' wellbeing; showed a positive correlation between the ease of studying online with the perceived efficiency of the university; showed that a positive perception of the university's efficiency will decrease the levels of stress and anxiety in students and also that migrating from the traditional to the online learning system caused students to have negative perceptions regarding their personal development and wellbeing. The study revealed that students experienced various situations and feelings that affected their wellbeing.

Gavriluta et al. conducted, between 20 April and 10 May 2020, a questionnaire-based survey among students (1013 respondents) from all Romanian universities. The research analyzed the educational, emotional, and social impact of the emergency state imposed by the COVID-19 pandemic on students and concluded that students accepted online education only as a form of compromise due to the epidemiological situation [30].

A study conducted by Cotoranu et al. during March and April 2021 on 321 students of Babeș-Bolyai University of Cluj-Napoca (Romania) revealed a negative psychological impact of the COVID-19 pandemic on most respondents regardless of the specialization in which they are enrolled (an average level of anxiety among respondents). Results indicated a high level of stress, nervousness, and difficulties in controlling the situation among students and showed that undergraduate students are more affected, younger students have a higher level of anxiety, and female students tend to be slightly more anxious than male students. The study also identified the main problems caused by the COVID-19 pandemic on students, such as monotony, boredom, sleep disorders, nervousness, agitation, and difficulties with concentration and motivation [31].

In October 2021, Săseanu et al. conducted a study that explored the effectiveness of online learning among Romanian students from the Bucharest University of Economic Studies during the COVID-19 pandemic. Based on a sample of 1952 respondents, the study reveals a significant difference in the students' perceptions of the online teaching activity depending on the domain of study, level of study, year of the study, digital skills, and internet connectivity [32].

Another research on 1415 students from five major Romanian faculties of economics, which tried to identify the determinants of effective online learning in the emergency situation created by the COVID-19 pandemic, concluded that psychological distress and increased concerns due to the pandemic situation had a negative effect on learning effectiveness. The results of the study also show that the effectiveness of online learning is influenced by several determinants: student gender, student age, student living area, internet connectivity, family issues, perception of the importance of the professor in the e-learning process, and eLearning framework of universities [33].

## 3. Research Methodology

The target audience was represented by students of first (bachelor's level) and second cycle (master's level) from six Romanian universities (Dunarea de Jos University of Galati, University of Agronomic Sciences and Veterinary Medicine of Bucharest, Politehnica University of Bucharest, National University of Arts George Enescu, Technical University Gheorghe Asachi Iasi, University of Bucharest) where approximative 40,000 students (bachelor and master's degree) are enrolled.

The main research tool was a questionnaire that was designed in order to determine the students' perceptions toward online education. The questionnaire was structured in three sections. The first section included a filter question in order to exclude the respondents that did not meet the inclusion criteria. The second section was designed to determine the students' perceptions toward online education, while the third section approached the respondents' profile (segmentation criteria): gender, residence, level, and field of study. A pilot survey was previously tested and validated on a small group of 20 students who were not included in the sample. The questionnaire, designed on Google Forms, had an introductory letter that presented the research objectives, informed respondents that no personal data were involved, and a commitment of the researchers to respect

the confidentiality of the answers and to release the answers only as a part of group summaries. The questionnaire was sent to the students using different communication channels (email, WhatsApp, and Teams), trying to reach as many students as possible. Data collection was achieved from 14 February 2022 to 14 March 2022, was supported by student associations, and resulted in a convenience sample of 388 respondents. The distribution of the population from our research sample according to the segmentation criteria reveals that the majority of respondents are females (69.3%), while the males represent 30.7%; the residence of the respondents is 66.2% urban and 33.8% rural; the education level shows that 67.01% of the respondents are bachelor's level and 32.99% are master's level. Regarding the respondents' field of study, 78.1% are from social sciences, 2.3% from biological science, 7% from engineering science, 9.5% from natural sciences, and 3.1% from humanities.

In this research, seven dependent variables were identified, such as teacher–student communication, student–teacher, student–institution (PS), logistical support from the educational institution (AL), adaptation according to knowledge economy requirements (EC), online teaching–learning activity in higher education (API), hybrid education model (eh), digital skills, and integration of digital technology in institutions (ITE) and resilient education system in the context of the COVID-19 pandemic (SER). A multi-item, five-point, bipolar Likert scale that ranged from "total disagreement" (1) to "total agreement" (5), for all indicators were selected. The item ratings were summarized to form a summated rating scale for each independent variable. Furthermore, since this is the first study of its kind within Exploratory Testing, all the items were written specifically for this study.

In order to adequately meet the specific objectives of quantitative research, the conceptual model was divided into three sub-models. The first sub-model highlighted the links between the seven latent variables and was designed according to the SEM-PLS method (Figure 1). Previous studies highlight the advantages of PLS-SEM, such as the possibility to estimate complex and innovative models and the method's flexibility in terms of data requirements and measurement specifications. Researchers from different fields use PLS-SEM for data analysis in their studies [34,35]. We choose PLS-SEM because structural equation modeling allows estimating cause-effect relationships between latent variables.

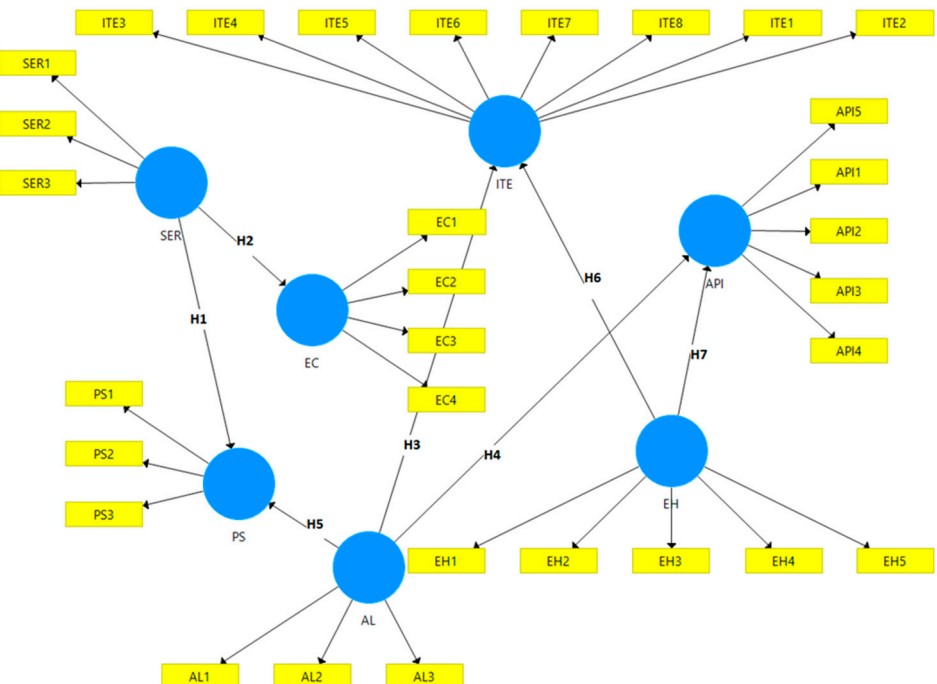

**Figure 1.** Conceptual sub-model approached using the SEM-PLS method.

The seven latent variables are reflective, as we used 3,4,5, or 8 items in the questionnaire to highlight the content of each one. The resilient education system in the context

of the COVID-19 pandemic (SER), teacher–student, student–teacher, student–institution communication (PS), and logistical assistance from the educational institution (AL) are characterized by three items, and adaptation according to knowledge economy requirements (EC) through four items, online teaching–learning activity in higher education (API), and hybrid education model (EH) are characterized by five items each, while digital skills and integration of digital technology in educational institutions (ITE) through eight items. Modeling using structural equations, using the least squares method (SEM PLS), gave us the opportunity to configure and estimate complex relationships between latent variables in this sub-model.

In the open context of education, we identify educational resilience as the ability to achieve school performance and the ability to cope with the challenges and pressures of the university school environment in the conditions of the crisis triggered by the COVID-19 pandemic. In our study, the system of resilient education in the context of the COVID-19 pandemic (SER) is defined both as pedagogical, individual resilience, capitalized through the student–teacher relationship, the student–colleague relationship, and the school-community relationship, but also as collective resilience through the initiative and effective action of the institutional management to provide technical support.

The seven hypotheses related to this sub-model are:

**Hypothesis 1.** *The resilient education system in the context of the COVID-19 pandemic (SER) has a significant effect on teacher–student, student–teacher, student–institution communication (PS).*

**Hypothesis 2.** *The resilient education system in the context of the COVID-19 pandemic (SER) has a significant effect on the adaptation according to knowledge economy requirements (EC).*

**Hypothesis 3.** *Logistical assistance from the educational institution (LA) has a significant effect on digital skills and the integration of digital technology in educational institutions (ITE).*

**Hypothesis 4.** *Logistical assistance from the educational institution (LA) has a significant effect on the teaching–learning activity (API).*

**Hypothesis 5.** *Logistical assistance from the educational institution (LA) has a significant effect on teacher–student, student–teacher, student–institution (PS) communication.*

**Hypothesis 6.** *The hybrid education (EH) model has a significant effect on digital skills and the integration of digital technology in educational institutions (ITE).*

**Hypothesis 7.** *The hybrid education (EH) model has a significant effect on teaching–learning activity (API).*

## 4. Results

The structural model highlights that the resilient education system in the context of the COVID-19 pandemic (SER) has the strongest effect on adapting to the requirements of the knowledge economy (EC), as the effect coefficient associated with this link is the highest (0.350). We also noticed that the hybrid education model (EH) has the weakest effect on digital skills and the integration of digital technology in educational institutions (ITE) coefficient of effect of only 0.143. Figure 2 illustrates the effect relationships between the latent variables included in the research sub-model approached by the SEM-PLS method, indicated by arrows oriented from latent variables considered independent to latent dependent variables.

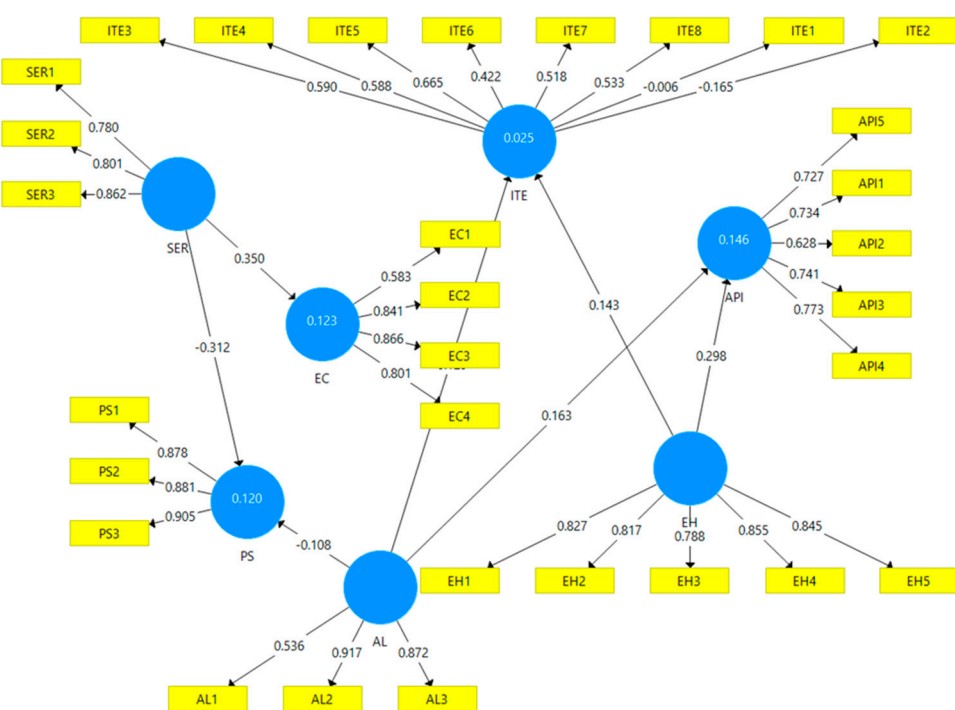

**Figure 2.** Illustration of effect coefficients and external loads of latent reflective variables.

Regarding outer loadings of the reflective latent variables, which reflect the statistical contributions of each item to each latent variable, we note:

- SER3 Item (Resilient Education System in the Context of the COVID-19 Pandemic) has the most representative statistical contribution to the latent variable (external load of 0.862, higher than in the case of SER1 and SER2);
- AL2 Item (Logistical assistance from the educational institution) has the most representative statistical contribution to the latent variable (external load of 0.917, higher than in the case of AL1 and AL3);
- PS3 item (Teacher–student, student–teacher, student–institution communication in online education) has the most representative statistical contribution to the latent variable (external load of 0.905, higher than in the case of PS2 and PS1);
- EC3 Item (Adaptation to the requirements of the knowledge economy) has the most representative statistical contribution to the latent variable (external load of 0.866, higher than in the case of EC1, EC2, EC4);
- EH4 Item (Hybrid education model) has the most representative statistical contribution to the latent variable (external load of 0.855, higher than in the case of EH1, EH2, and EH3);
- API4 item (Teaching–learning activity in online education) has the most representative statistical contribution to the latent variable (external load of 0.773, higher than in the case of API1, API2, API3, and API5);
- ITE5 item (Digital competencies and the integration of digital technology in educational institutions) has the most representative statistical contribution to the latent variable (external load of 0.665, higher than in the case of ITE1, ITE2, ITE3, ITE4, ITE6, ITE7, and ITE8).

The evaluation of the measurement sub-model based on the modeling of the seven structural equations will be done by determining the level of internal consistency (SmartPLS software will calculate Cronbach Alpha and composite confidence level), convergent validity (SmartPLS software will generate a variance report extracted media) and discriminant validity (SmartPLS software will generate reports on the Fornell–Larcker criterion and the Heterotrait–Monotrait ratio (HTMT)) [36]. The variance inflation factor (VIF) that measures the multicollinearity of a set of variables in a multiple regression will not be

calculated, as all variables reflect a reflective approach, and this factor is calculated only for the formative approaches of latent variables in SEM PLS logic.

The Cronbach Alpha indicator highlights the internal consistency and implicitly the reliability of the research tool, as well as the correlation between the latent variables integrated with the structural sub-model. The minimum threshold accepted by statisticians for this indicator is 0.7. Cronbach Alpha values exceed the allowable threshold for the variables AL (0.701), API (0.770), EC (0.776), PS (0.866), SER (0.747), and EH (0.884), while for the variable ITE, they are below near the minimum allowable threshold: 0.469).

We kept this variable in our sub-model because it increases AVE and CR values for the other variables.

Convergent validity refers to 'the extent to which a measure(s) positively correlates with alternative measures (indicators) of the same construct'. Table 1 shows the details of each construct used in the conceptual framework, and the indicators associated with each construct are listed in the column 'element'. First, composite reliability is checked, and any value below the 0.7 limit value should be assessed. This is established by checking the reliability values of the indicator.

**Table 1.** Assessment of internal consistency and convergent validity within the evaluated sub-model.

| | Construct Reliability and Validity | | | |
| --- | --- | --- | --- | --- |
| | **Cronbach's Alpha** | **rho_A** | **Composite Reliability** | **Average Variance Extracted (AVE)** |
| **AL** | 0.701 | 0.824 | 0.830 | 0.630 |
| **API** | 0.770 | 0.779 | 0.844 | 0.521 |
| **EC** | 0.776 | 0.786 | 0.860 | 0.610 |
| **EH** | 0.884 | 0.886 | 0.915 | 0.684 |
| **ITE** | 0.469 | 0.551 | 0.618 | 0.237 |
| **PS** | 0.866 | 0.869 | 0.918 | 0.788 |
| **SER** | 0.747 | 0.752 | 0.856 | 0.665 |

The composite confidence level considers the variable loads of all indicators, being more flexible in this respect than Cronbach Alpha. The minimum allowable threshold for the composite confidence level is also 0.7, and in the case of our research, six variables exceed it. Spearman's rank correlation coefficient (Rho) is a nonparametric test whose values fall between –1 and = 1. The value r = 1 reflects a perfect positive correlation, and the value r = –1 is associated with a perfect negative correlation. Note that in the case of the seven reflective variables, six have positive correlations.

The convergent validity of the sub-model is determined by the average extracted variance (AVE), which measures the variance of a latent variable relative to the variance associated with the measurement error. In general, statisticians recommend a minimum AVE threshold of 0.5. We note that six variables (AL, API, EC, EH, PS, and SER) have values of the mean variance extracted above the recommended threshold, which validates the convergent validity of this sub-model for measuring the relationships between variables.

To determine the discriminant validity, we will first apply the Fornell–Larcker criterion, which compares the square root of the extracted average variance (AVE) with the correlation of latent variables. Statisticians recommend that the square root of the AVE of each reflective variable be greater than the correlations with other latent variables, a fact confirmed in this empirical research (since the AVE values for AL (0.793), API (0.722), EC (0.781), EH (0.827), PS (0.888) and SER (0.815) are superior to the correlations with the other latent variables, positioned below the main diagonal in Table 2.

**Table 2.** Assessment of discriminant validity in the case of the analyzed sub-model (Fornell–Larcker criterion).

| | Fornell–Larcker Criterion | | | | | | |
|---|---|---|---|---|---|---|---|
| | **AL** | **API** | **EC** | **EH** | **ITE** | **PS** | **SER** |
| **AL** | 0.793 | | | | | | |
| **API** | 0.258 | 0.722 | | | | | |
| **EC** | 0.364 | 0.347 | 0.781 | | | | |
| **EH** | 0.320 | 0.350 | 0.502 | 0.827 | | | |
| **ITE** | −0.084 | −0.047 | −0.018 | 0.102 | 0.487 | | |
| **PS** | −0.158 | −0.259 | −0.417 | −0.386 | −0.132 | 0.888 | |
| **SER** | 0.160 | 0.216 | 0.350 | 0.400 | 0.049 | −0.329 | 0.815 |

The second way to determine discriminant validity is provided by the Heterotrait–Monotrait (HTMT) correlation report. HTMT is considered by statisticians to be more appropriate for assessing discriminant validity than the Fornell–Lacker criterion in terms of superior performance, which allows it to achieve higher reliability rates [37]. HTMT values approaching the maximum allowable threshold of one indicate discriminatory invalidity. The use of HTMT as a criterion implies its comparison with a predefined maximum threshold indicating the existence of discriminant validity, considered by most researchers to be 0.9. In the case of this research, we observe that only the correlation EH-> EC between all variables slightly exceeds the level of 0.6, well below the maximum threshold of 0.9. Thus, the discriminant validity of the sub-model is also validated by this criterion (Figure 3).

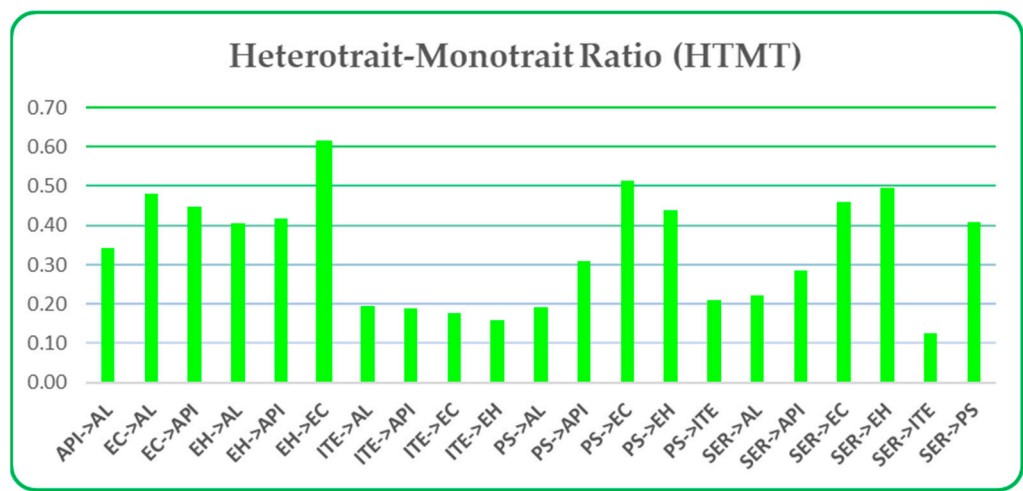

**Figure 3.** Application of the HTMT report to assess discriminatory validity.

The PLS-SEM method focuses on the principle that the data do not have normalized statistical distributions, which requires the application of a bootstrapping procedure to allow the running of significance tests between the hypotheses of the sub-model. Through the bootstrapping procedure, subsamples are created with observations randomly extracted from the original data set (by successive replacements), which are used to estimate the new structural model.

Estimates of the parameters associated with the analyzed structural sub-model (external variable loads and estimated relationship coefficients in the subsamples) are used to generate statistical reports, which reflect *t*-test values and asymptotic meanings (*p* values). These statistical tests are able to validate and invalidate the hypotheses of the sub-model. Figure 4 reflects the new structural model generated by the SmartPLS software after apply-

ing the bootstrapping procedure; we notice that on the links between the latent variables, the *p* values related to the asymptotic significance are highlighted.

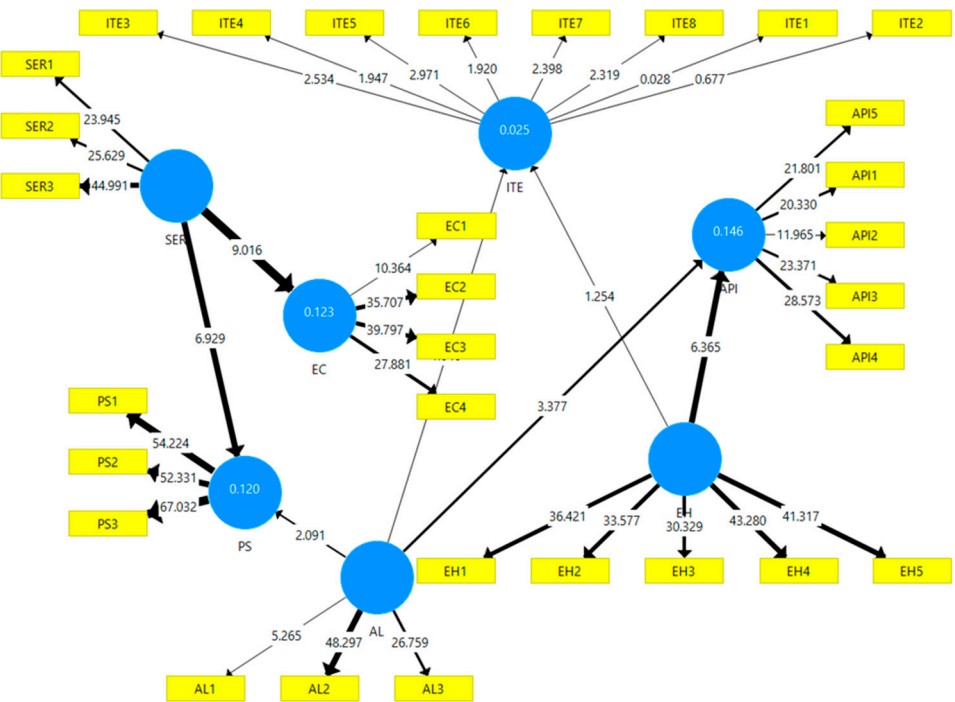

**Figure 4.** Determining the *p* values associated with the relationships between the variables of the sub-model.

Of the seven hypotheses of this sub-model, only two are not validated, as the *p* values exceed the maximum allowed significance level of 0.05, namely (Table 3):

**Hypothesis 3:** Logistical assistance from the educational institution (AL) has a significant effect on digital skills and the integration of digital technology in educational institutions (ITE)—asymptotic significance value *p* = 0.092.

**Hypothesis 6:** The hybrid education (EH) model has a significant effect on digital skills and the integration of digital technology in educational institutions (ITE)—asymptotic significance value *p* = 0.182.

**Table 3.** The values associated with the asymptotic significance *p* and the *t*-test for the 7 hypotheses in the structural sub-model.

| | Mean, STDEV, *t* Values, *p* Values | | | | |
|---|---|---|---|---|---|
| | Original Sample (O) | Sample Mean (M) | Standard Deviation (STDEV) | *t* Statistics (⏐O/STDEV⏐) | *p* Values |
| **AL->API** | 0.163 | 0.166 | 0.049 | 3.294 | 0.001 |
| **AL->ITE** | −0.129 | −0.154 | 0.077 | 2.101 | 0.092 |
| **AL->PS** | −0.108 | −0.108 | 0.051 | 1.996 | 0.036 |
| **EH->API** | 0.298 | 0.305 | 0.044 | 6.710 | 0.000 |
| **EH->ITE** | 0.143 | 0.154 | 0.107 | 1.338 | 0.182 |
| **SER->EC** | 0.350 | 0.355 | 0.042 | 8.390 | 0.000 |
| **SER->PS** | −0.312 | −0.317 | 0.047 | 6.587 | 0.000 |

The *t*-test shows the strength of the correlation between the latent variables in this structural sub-model. Thus, the resilient education system in the context of the COVID-19 pandemic (SER) has a significant effect on the adaptation to the requirements of the knowledge economy (EC)—*t* value = 8.390.

## 5. Discussions

First of all, we have to admit that Romania ranked 26th out of the 28 EU countries regarding digitalization. The degree of PC coverage of the students in the Romanian higher education system in the academic year 2019–2020 was approximately 19% [38]. The sudden and unplanned transition to online education has created challenges for students and teachers, who have had to adapt to the teaching and learning process in the new organizational and communication framework.

During the COVID-19 pandemic, the online learning infrastructure was developed, and students from disadvantaged backgrounds received devices from the educational institution to attend online classes. The impact of this measure caused a low university dropout.

Based on the data analysis, we concluded that concentration difficulties during school activities were encountered by students who were in a family setting. These difficulties were mentioned in a similar study; the students surveyed were mentioned as being more likely to be distracted during classes by a family member or household chores, but also by video games and social media [39,40]. Concentration difficulties negatively affect students' self-confidence. This situation may also be due to the lack of direct social interaction between students and teachers, respectively student–students, but also to the long period of time they have to spend in front of the laptop / PC, which inevitably leads to loss of concentration and the installation of boredom and monotony with a negative impact on their academic performance [41].

Authentic assessments and timely feedback are key components of learning. A very important part of online learning is the availability of useful formative assessments and timely feedback for students in online education [42]. This continues to be a challenge for educators and the education system. At the same time, the students participating in the survey described several barriers related to digital skills and the integration of digital technology in educational institutions, the most representative barrier being the lack of internet/poor internet connection. This is similar to recent findings regarding the cancellation of face-to-face teaching, where teaching staff and students were forced to work and learn from home during the closure of universities [43].

Our research highlights a significant correlation between logistical assistance from the educational institution (LA) and teacher–student, student–teacher, student–institution (PS) communication. Moreover, logistical assistance from the educational institution (LA) has a significant effect on the teaching–learning activity. Similar results are obtained by researchers that, using questionnaire analyses with a sample of 1415 students from Romania, highlight the link between logistical assistance and the efficiency of teaching–learning activity [41]. The involvement of educational institutions in providing logistical support during the COVID-19 pandemic diminished the effects of education process changes.

The results of previous research show that students react differently to online classes; their reaction is based on their abilities to use online tools. Furthermore, most students consider online classes stressful and prefer online exams. Students' dissatisfaction with online courses suggests the lack of attractiveness and interactivity of these courses and especially the overcrowding of the program [43]. We also obtained similar results, as the hybrid education (EH) model has a significant effect on teaching–learning activity (API). Another important finding of our study shows that 86% of respondents want to keep the online system for different activities such as homework, exams, meetings, and communication with professors. The great majority of students have an open attitude toward the use of ICT tools in their current activities as an effect of online education. This result is similar to other recent findings on the crucial role of e-learning tools during the pandemic [44].

In our research, we prove that a hybrid model of education has a positive effect on digital skills and the integration of digital technology in educational institutions. The COVID-19 pandemic has given us the opportunity to pave the way for the introduction of digital learning.

Like all empirical studies, this research has several limitations. The respondents who formed our sample do not represent all the university centers. Moreover, our study participants are not students in all fields and nationally recognized undergraduate and master's degree programs. Almost 80% of respondents are from the social science field; therefore, we cannot generalize the conclusions of our research for all fields of study.

In the future, we intend to extend the analysis by including in the research model other variables that influence the educational process.

## 6. Conclusions

The SARS-CoV-2 pandemic has had a dramatic impact on the world. However, as with all other crises, in addition to the negative effects, the positive effects must be highlighted. Education can benefit from this digital transformation. Online education can be especially useful not only for formal education but also for informal education, long-term education, and for most of the staff active in higher education institutions [45]. We must consider that school is not only a space for academic learning but also for the development of social and emotional skills. Modeling using structural equations, using the least partial squares method (SEM PLS), gave us the opportunity to configure and estimate complex relationships between latent variables.

Of the seven hypotheses, two are not validated, as the $p$ values exceed the maximum allowed significance level of 0.05. Thus, Logistical assistance from the educational institution (AL) has a significant effect on digital skills and the integration of digital technology in educational institutions (ITE)—asymptotic significance value $p = 0.092$ (H3)—and the hybrid education (EH) model has a significant effect on digital skills and the integration of digital technology in educational institutions (ITE)—asymptotic significance value $p = 0.182$ (H6)

The other five hypotheses are validated, and our research highlights that the hybrid education (EH) model has a significant effect on digital skills and the integration of digital technology in educational institutions (ITE). Moreover, the resilient education system in the context of the COVID-19 pandemic (SER) has a significant effect on the adaptation to the requirements of the knowledge economy (EC) and on the teacher–student, student–teacher, student–institution communication (PS).

If the assumptions used were to become certain, then the Romanian education system in general, and the university system in particular, would be able to ensure a reduction in the effects generated by unexpected events, such as the COVID-19 pandemic, to ensure a higher degree of preparedness for possible future crises and to contribute significantly to the improvement of quality inclusive education. In this context, the Romanian Ministry of Education has made a legislative proposal to extend online education in universities, especially for master's programs.

**Author Contributions:** Conceptualization, S.D. and L.D.M.; methodology, S.D. and F.O.V., N.B.-M.; validation, L.D.M., I.A.Ș. and S.D.; formal analysis, F.O.V. and N.B.-M.; investigation, N.B.-M.; resources, L.D.M.; data curation, I.A.Ș.; writing—original draft preparation, F.O.V.; visualization, I.A.Ș.; All authors have read and agreed to the published version of the manuscript.

**Funding:** This research was funded by "Dunarea de Jos" University of Galati, Research Grant.

**Institutional Review Board Statement:** Not applicable.

**Informed Consent Statement:** Informed consent was obtained from all subjects involved in the study.

**Data Availability Statement:** The data presented in this study are available from the corresponding author upon reasonable request.

**Conflicts of Interest:** The authors declare no conflict of interest.

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
