# Peer review of "Higher Education Institution beyond the COVID-19 Pandemic—Evidence from Romania"

_education, doi:10.3390/educsci12100693_

Round 1
Reviewer 1 Report
This study is based on a questionnaire analysis, with 388 respondents from 6 different universities. The author/s used the SEM-PLS method to verify a conceptual model. The proposed model is based on 7 latent variables. It may belong to a complex model to be tested. Some suggestions for revision are listed as follows.
1. The introduction section may need clear contextual impacts and research questions to explore.
2. In the literature section, it may need more work, for example, clarity of context for Romanian universities during COVID-19; or the issue extended to Asia or global settings. The related literature needs to be cited properly.
3. Some citations are not correct in the proper position, for example, line 43 [4]; line 54 [11,12]; line 73, 79…..please make it consistent.
4. In the method section, please define your observed items. Or if the observed items are designed by yourself or cited from literature, you need to clarify.
5. In Table 1, the assessment of internal consistency and convergent validity, the value of Cronbach's Alpha and AVE are related to low in ITE, why?
6. Please provide a section to address this study's related limitations.
Author Response
R1
This study is based on a questionnaire analysis, with 388 respondents from 6 different universities. The author/s used the SEM-PLS method to verify a conceptual model. The proposed model is based on 7 latent variables. It may belong to a complex model to be tested. Some suggestions for revision are listed as follows.
- The introduction section may need clear contextual impacts and research questions to explore.
Authors: We improved this section.
The research questions formulated for the study are as follows:
RQ1: What are the challenges in higher education institutions beyond the Covid-19 pandemic from students’ perception?
RQ2: How is the students’ perspective on the hybrid education model?
RQ3: What is the relationship between the latent variables involved in the research?
Thank you for your suggestions!
- In the literature section, it may need more work, for example, clarity of context for Romanian universities during COVID-19; or the issue extended to Asia or global settings. The related literature needs to be cited properly.
Authors: We completed with relevant studies regarding Romanian higher education system in the pandemic context . Thank you!
- Some citations are not correct in the proper position, for example, line 43 [4]; line 54 [11,12]; line 73, 79…..please make it consistent.
Authors: We corrected. Thank you for your suggestions!
- In the method section, please define your observed items. Or if the observed items are designed by yourself or cited from literature, you need to clarify.
Authors: For this research study, seven dependent variables were identified, such as teacher-student communication, student-teacher, student-institution (PS), logistical support from the educational institution (AL), adaptation to the requirements of the knowledge economy (EC), teaching-learning activity in online education (API), hybrid education model (eh), Digital skills and integration of digital technology in institutions (ITE) and resilient education system in the context of the Covid pandemic (SER). A multi-item, five-point, bipolar Likert scales that ranged from “total disagreement” (1) to “total agreement” (5) for all indicators were selected. The item ratings were summarized to form a summated rating scale for each independent variable. Furthermore, since this is the first study of its kind within Exploratory Testing, all the items were written specifically for this study.
- In Table 1, the assessment of internal consistency and convergent validity, the value of Cronbach's Alpha and AVE are related to low in ITE, why?
Authors: The Cronbach Alpha indicator highlights the internal consistency and implicitly the reliability of the research tool, as well as the correlation between the latent variables integrated in the structural sub-model. The minimum threshold accepted by statisticians for this indicator is 0.7. Cronbach Alpha values exceed the allowable threshold for the variables AL (0.701), API (0.770), EC (0.776), PS (0.866), SER (0.747) and EH (0.884), while for the variable ITE, they are below near the minimum allowable threshold: 0.469). We kept this variable in our sub-model because it increases ave and CR values for the other variables.
Convergent validity refers to ‘the extent to which a measure(s) positively correlates with alternative measures (indicators) of the same construct’. Table 1 shows the details of each construct used in the conceptual framework and the indicators associated with each construct are listed in the column ‘element’. First, composite reliability is checked and any value below the 0.7 limit values should be assessed. This is established by checking the reliability values of the indicator.
- Please provide a section to address this study's related limitations.
Authors: The limitations of our study are highlighted in the discussion section.
We tried to make the necessary corrections in line with your comments and suggestions. We think you have greatly improved our work. Thank you!
Reviewer 2 Report
Introduction and Literature Review are very thorough, although some English grammar mistakes decreased the readability of this manuscript.
Research Methods are lacking citations. Also, subsections or more well constructed paragraphs would increase understanding of the methods. Describe and cite the research design, population and sample, instrumentation (including validity and reliability analyses), data collection procedures, and data analyses. We don't know anything about the instrument; therefore, it may be invalid for this study. How was it developed? Who developed it? Too much missing information. "Finally, 388 questionnaires were filled." These results represent ONLY those who responded; there was no sample, nor population.
MUST tone down language that infers to a larger population, or "students" in general because the results are not from a random or even a purposive sample.
"Latent variables" should be described in relationship to the literature reviewed, otherwise there's no connection between previous studies and this study.
Results are missing statistical notation in the narrative. Cronbach alpha analyses should be part of the Methods, not buried in Results. The same is true for determining discriminant validity. Also, more focused writing is needed. This is not a study about instrument validation (because we do not know anything about the actual instrument), but is a study about 388 students' perceptions of the effects of COVID-19 on learning in Romania.
The authors should be recognized for this initial attempt to report findings from the effects of COVID-19 on education in Romania. Some of the writing loses focus, is missing important information, and has logic gaps. These errors can be remedied with a well stated research problem, purpose of research, and objectives. Those elements help writers maintain focus throughout the research process.
Author Response
We improved the Introduction and Literature Review. Also, we insert the citation on the Research Methods (bibliographic indices 35,36).
For this research study, seven dependent variables were identified, such as teacher-student communication, student-teacher, student-institution (PS), logistical support from the educational institution (AL), adaptation to the requirements of the knowledge economy (EC), teaching-learning activity in online education (API), hybrid education model (eh), Digital skills and integration of digital technology in institutions (ITE) and resilient education system in the context of the Covid pandemic (SER). A multi-item, five-point, bipolar Likert scales that ranged from “total disagreement” (1) to “total agreement” (5) for all indicators were selected. The item ratings were summarized to form a summated rating scale for each independent variable. Furthermore, since this is the first study of its kind within Exploratory Testing, all the items were written specifically for this study.
The Cronbach Alpha indicator highlights the internal consistency and implicitly the reliability of the research tool, as well as the correlation between the latent variables integrated in the structural sub-model. The minimum threshold accepted by statisticians for this indicator is 0.7. Cronbach Alpha values exceed the allowable threshold for the variables AL (0.701), API (0.770), EC (0.776), PS (0.866), SER (0.747) and EH (0.884), while for the variable ITE, they are below near the minimum allowable threshold: 0.469). We kept this variable in our sub-model because it increases ave and CR values for the other variables.
Convergent validity refers to ‘the extent to which a measure(s) positively correlates with alternative measures (indicators) of the same construct’. Table 1 shows the details of each construct used in the conceptual framework and the indicators associated with each construct are listed in the column ‘element’. First, composite reliability is checked and any value below the 0.7 limit values should be assessed. This is established by checking the reliability values of the indicator.
We tried to make all necessary corrections in line with your comments and suggestions.
We think you have greatly improved our work. Thank you!
Reviewer 3 Report
Topic of the article Higher Education Institution beyond the COVID-19 Pandemic. Evidence from Romania is interesting and current. In the article, the authors address aspects of higher education in Romania in the context of the COVID-19 pandemic and online education. The authors describe their own research on a sample of approximately 380 students from 6 universities. The authors established 7 hypotheses for which they sought answers using the SEM-PLS method. H1 and H2 hypothesis refer to the concept of "The resilient education system", but it would be appropriate to define this term more closely. The other hypotheses are clear. The research results are clearly presented and I think that the results are also applicable to other countries. I recommend accepting it after completing the time horizon when the questionnaire research was carried out and clarifying the term "The resilient education system".
Author Response
We completed the time horizon when the questionnaire research was carried out and we define the term "The resilient education system".
In the open context of education, we identify educational resilience as the ability to achieve school performance and the ability to cope with the challenges and pressures of the university school environment in the conditions of the crisis triggered by the COVID -19 pandemic. In our study, the system of resilient education in the context of the Covid pandemic (SER) is defined both as pedagogical, individual resilience, capitalized through the student-teacher relationship, the student-colleague relationship and the school-community relationship, but also as collective resilience through the initiative and effective action of the institutional management to provide technical support.
We tried to make the necessary corrections in line with your comments and suggestions. We think you have greatly improved our work. Thank you!

Round 2
Reviewer 2 Report
Thank you for responding to the original review and providing clarity on several issues. While many items were corrected, I did not revisions in descriptions of the "target population" or "sample." How many students were enrolled in the six Romanian universities? A sample is mentioned, but not what type of sample. Was it a random sample of all students at the six universities? Was it a proportionally-stratified random sample? Was it a convenient sample? A few statements about sample type and rationale for sampling methods would provide much clarity.
Considering the limitations of the study, some of the language is still overgeneralized to audiences beyond the limits of the study.
Author Response
Authors:
The target population was represented by students of first (bachelor’s level) and second cycle (master’s level) from six Romanian universities (Dunarea de Jos University of Galati, University of Agronomic Sciences and Veterinary Medicine of Bucharest, Politehnica University of Bucharest, National University of Arts George Enescu, Technical University Gheorghe Asachi Iasi, University of Bucharest), where approximative 40000 students (bachelor and master’s degree) are enrolled.
We distributed the questionnaire online, in some universities benefiting from the support of the student associations. We mention in the research method section, that it was a convenience sample. This sample method doesn’t require a random selection of participants based on any set of criteria. We know that convenience sample has important disadvantages (sampling bias, difficult to generalize the data, low external validity).
We consider as a limit of our article the difficulty to generalize the conclusions of the research study.
Thank you for your comments!